

# Psychometric evaluation of a newly developed Elderly-Constipation Impact Scale

Patimah Abdul Wahab[1,2], Dariah Mohd Yusoff[2], Azidah Abdul Kadir[3],
Siti Hawa Ali[4], Yeong Yeh Lee[5,6,7] and Yee Cheng Kueh[8]

[1] Department of Medical-Surgical Nursing, Kulliyyah of Nursing, International Islamic University Malaysia, Kuantan, Malaysia
[2] Nursing Programme, School of Health Sciences, Universiti Sains Malaysia, Kota Bharu, Malaysia
[3] Department of Family Medicine, School of Medical Sciences, Universiti Sains Malaysia, Kota Bharu, Malaysia
[4] Unit of Interdisciplinary Health Sciences, School of Health Sciences, Universiti Sains Malaysia, Kota Bharu, Malaysia
[5] School of Medical Sciences, Universiti Sains Malaysia, Kota Bharu, Malaysia
[6] Gut Research Group, Faculty of Medicine, Universiti Kebangsaan Malaysia, Kuala Lumpur, Malaysia
[7] St George & Sutherland Clinical School, University of New South Wales, Sydney, Australia
[8] Unit of Biostatistics and Research Methodology, School of Medical Sciences, Universiti Sains Malaysia, Kota Bharu, Malaysia

Corresponding author
Patimah Abdul Wahab,
patimah@iium.edu.my

## ABSTRACT

**Background**. Chronic constipation is a common symptom among the elderly, and it may affect their quality of life (QoL). A lack of available research focused on the elderly means that this effect is not well understood. This study aimed to develop and validate a new scale (Elderly-Constipation Impact Scale (E-CIS)) to measure the impact of chronic constipation on QoL among the elderly.

**Methods**. A pool of items was generated from a qualitative study, literature reviews, and expert reviews. Exploratory factor analysis (EFA) was performed on the original 40 items of the E-CIS and followed by 27 items for confirmatory factor analysis (CFA). A total of 470 elderly people with chronic constipation were involved.

**Results**. The mean age of the participants was $68.64 \pm 6.57$. Finally, only 22 items were indicated as appropriately representing the E-CIS, which were grouped into seven subscales: 'daily activities', 'treatment satisfaction', 'lack of control of bodily function', 'diet restriction', 'symptom intensity', 'anxiety' and 'preventive actions'. The scale was confirmed as valid (root mean square error of approximation (RMSEA) = 0.04, comparative fit index (CFI) = 0.961, Tucker-Lewis index (TLI) = 0.952 and chi-square/degree of freedom (chiSq/df) = 1.44) and reliable (Cronbach's alpha: 0.66–0.85, composite reliability (CR) = 0.699–0.851) to assess the impact of chronic constipation on the elderly's QoL.

**Conclusions**. The E-CIS is useful to measure the impact of chronic constipation on the elderly's QoL. A further test is needed to determine the validity and reliability of this scale in other elderly population.

## INTRODUCTION

Constipation occurs frequently among the elderly and its prevalence may be as high as 20% of community-dwelling elderly people and 74% of nursing home residents (*Bouras & Tangalos, 2009*). It is a symptom-based disorder of the gastrointestinal system which refers to a condition of having difficulties in bowel opening or infrequent passage of stool, hard stool, or a feeling of incomplete evacuation (*Bharucha, Pemberton & Locke, 2013*). Previous studies showed that constipation may cause a considerable impact on quality of life (QoL) and functional status, as well as being a significant driver of healthcare costs (*Dennison et al., 2005*; *Nellesen et al., 2013*; *Pinto Sanchez & Bercik, 2011*; *Shalmani et al., 2011*; *Sun et al., 2011*). The impact of constipation on QoL is usually predominated by mental health, and the magnitude of the impact is comparable to other diseases (*Belsey et al., 2010*). This impact is more obvious among the elderly with persistent or chronic conditions and the presence of abdominal pain or discomfort (*Johanson & Kralstein, 2007*; *Jun et al., 2006*; *Koloski et al., 2013*).

The impact of constipation on the elderly's QoL is usually measured using constipation-specific of generic instruments. The use of generic instruments is limited in terms of comparing the impact on QoL of those with and without constipation, but the constipation-specific instrument allows the detection of the specific impact of constipation on QoL. So far, there is one main constipation QoL scale (Patient Assessment of Constipation Quality of Life (PAC-QoL)) with psychometric validity and reliability available for the elderly population (*Frank, Flynn & Rothman, 2001*; *Marquis et al., 2005*). However, its usage is not suitable for the elderly because the impact is measured based on five items under the satisfaction domain, and the scale was validated only on the feasibility of the interview format among 24 nursing home residents with chronic constipation.

The existing constipation QoL scales also differ in terms of the domains measured. For example, *Szeinbach et al. (2009)*, focused only on patients' satisfaction with the treatment because of its closed relationship with the other domains like daily life activity and psychosocial domains. In contrast to this, *Wang et al. (2009)*, argued that the satisfaction domain is more likely to measure symptom severity, while the physical domain is not measuring the impact of chronic constipation on daily life activity. Based on the revised Wilson and Cleary model of health-related quality of life (HRQoL), an individual's constipation-QoL is influenced by the individual's characteristics, environment, biological function, symptoms, functional status, general health perceptions and overall QoL (*Ferrans et al., 2005*; *Bakas et al., 2012*).

Patimah and colleagues (*2017*) emphasized that the existing constipation-QoL scales were developed in Western countries and, thus, they have lack of cultural sensitivity and reference to issues of age. Most of the domains and items measured such as work/leisure, psychosocial or social impairment, and treatment satisfaction refer to Western cultures and lifestyles which are unsuitable for other populations, particularly elderly residents of Asian countries. A good scale should be developed based on the importance of the domains measured to the study population and the impact of symptoms as defined by the patient (*Byrne et al., 2002*; *Damon, Dumas & Mion, 2004*; *Nilsson, Parker & Kabir, 2004*). Thus, this

study aimed to develop and validate a culturally sensitive scale—the Elderly Constipation Impact Scale (E-CIS)—with the purpose of assessing the impact of chronic constipation on QoL among elderly people, regardless of treatments they received or practised. In this study, we used the World Health Organization definition "a state of complete physical, mental, and social well-being not merely the absence of disease or infirmity" (*International Health Conference, 2002*) to define the impact of chronic constipation on health related QoL.

## MATERIALS & METHODS

### Sample

The sample of this study was elderly with chronic constipation living in community settings of states of Terengganu and Kelantan. They were defined based on Malaysia's National Policy for Older Persons, which is an individual age 60 years old and above (*Ministry of Women Family Community Development, 2011*). The criteria for inclusion were that participants must be men or women, who had lived in the community for at least 12 months and were able to understand Malay language (spoken in the local dialect). The criteria for exclusion were cognitive impairment—as determined by the elderly cognitive assessment questionnaire (ECAQ) (*Kua & Ko, 1992*; *Kua & Ko, 1995*), hospitalization, significant hearing impairment, stoma, and gastrointestinal cancers. These eligibility criteria were applied to participants for all phases.

For diagnosis of chronic constipation, the elderly participants were asked the following question: 'Do you have constipation for the past three months?', and, if they answered 'Yes', they were asked to rate the level of severity based on a 5-point Likert scale (1 = not severe to 5 = very severe). Respondents with constipation for the past three months, with a severity level of at least 'slightly severe', were invited to participate in the study. The aim was to involve a heterogeneous population that allowed for the detection of differences in scores across the new scale (*Marquis et al., 2005*; *Szeinbach et al., 2009*). General health status was measured by asking the participants to rate their current perception on health status using a 5-point Likert scale between poor and very good.

The participants were recruited from two community settings: Terengganu in Phase 2 and Kelantan in Phase 3 of the study. A random cluster sampling to eight districts of Terengganu resulted in the elderly living in community setting in the district of Marang were selected. In Kelantan, purposive sampling was used to recruit the participants by using the homogenous sampling method (*Etikan, Musa & Alkassim, 2016*). They were among those who attended a family medicine specialist clinic in a teaching hospital situated in Kelantan.

The total sample size of this study was 470 participants. In Phase 2, the sample size of 200 was derived based on a five participants per item ratio (*Anthoine et al., 2014*). Another group of elderly people were then recruited in Phase 3 with the sample size of 270, which was calculated based on the ratio of 10 participants to one item (*Fabrigar & Wegener, 2012*; *Nunnally, 1978*).

## Data assessment

The E-CIS was developed and validated in three phases: (a) items generation (Phase 1); (b) items reduction and establishment of factor structure (Phase 2), and; (c) confirmation of the factor structure and items (Phase 3). Meanwhile, its reliability was examined in the last two phases (Fig. 1). Findings of Phase 1 was used to generate initial items for the new scale. Following this, two cross-sectional studies were carried out among the elderly of a community setting in Terengganu and Kelantan from October 2015 to July 2016 to evaluate its psychometric property. A set of questionnaires consisting of socio-demographic data and items of the new scale were completed. In Phase 2, a face-to-face interview from door-to-door was conducted by an investigator and a research assistant (PAW & NFR) to prevent incomplete data. A similar approach of data collection was applied by the similar researcher in Phase 3 to avoid bias. The study involving human participants has obtained ethical approval from the Research Ethics Committee (Human), Universiti Sains Malaysia (USM/JEPeM/272.3. [1.7]). Permission to conduct the study was granted by the headmen and hospital director. Prior to data collection, verbal and written consents were obtained from the participants.

### Items generation

An initial pool of items was generated based on inputs from three sources that consisted of data from semi-structured interviews, comprehensive literature reviews, and opinions from an interdisciplinary group of experts. Data of semi-structured interviews were derived from a previously published study by the same group of investigators (*Patimah et al., 2017*). Initially, relevant codes from interviews were rereviewed to form items using layman descriptors in the standard Malay language. Crosschecking of these codes with literature reviews produced 74 items (*Marquis et al., 2005*; *Bowling & Stenner, 2011*).

To ensure content validity, all 74 items were reviewed by an expert panel that consisted of two nurses, two medical doctors, two lecturers, and one researcher (DMY, AFI, LYY, AAK, SHA, MSBY, & IIH). In addition, decisions to delete or to retain items were guided by the content validity index (CVI), an index based on the aggregated ratings of a panel of experts (*Polit & Beck, 2006*). The recommended minimum CVI for scale (S-CVI) is 0.80 and for individual items (I-CVI) is 0.78 (*Polit & Beck, 2006*; *Polit & Beck, 2012*). For E-CIS, the S-CVI was 0.77 and I-CVI ranged 0.40 to 1.00. Subsequently, items with low I-CVI were either removed or revised and, at the end of this process, 43 of 74 items remained. Next, the remaining 43 items and their responses (visual analogue scale and Likert scale) were checked for wording, content, appropriateness and convenience by two elderly people with chronic constipation (IM & RY).

The 43-item questionnaire was then pre-tested on another 20 elderly people with chronic constipation recruited via a face-to-face approach in an informal educational center. A debriefing interview was conducted, where they were asked to complete the questionnaire and they were then asked to explain their understanding of the items using their own words. They were also asked to comment and provide suggestions on the instructions and contents of the questionnaire. After this process, three items were removed, and 40 items were further examined for their factor structure.
**Phase 1: Items generation**
No. of items = 74
Temporary domains: 7

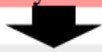

**Restructure:**
a) Content validity:
No. of items = 43
b) Face validity:
No. of items = 40

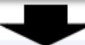

**Phase 2: Items reduction and establishment of factor structure**
Data analysis: exploratory factor analysis (EFA)

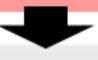

**Restructure:**
a) No. of items = 27
b) No. of factors = 8

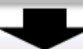

**Phase 3: Confirmation of the factor structure and items**
Data analysis: Confirmatory factor analysis (CFA)

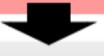

**Restructure:**
a) No. of items = 22
b) No. of factors = 7

**Figure 1** Development and validation of the scale.

To summarize, the first version of E-CIS has 40 items of 5-point Likert scale. Responses on impact are measured based on frequency of experience, where 1 = never or very rare and 5 = always. In these items, a higher score indicates a greater impact of chronic constipation on QoL of the elderly person. On the other hand, four items are rated based on the level of agreement to their treatment satisfaction, with values ranging from 1 = strongly disagree to 5 = strongly agree. These are reversed coded items. The total score is determined as a percentage by summating the scores per the total score and multiplying by 100.

### Item reduction and establishment of factor structure

Phase 2 was an exploratory factor analysis (EFA) to further reduce irrelevant items and to establish a factor structure. Principal axis factoring with promax rotation extraction was applied to explore the structure within the data as a basis for removal of redundancy

or unnecessary items. The number of factors was determined based on the scree plot, eigenvalues greater than 1.0, and that they met the variance criterion of higher than 60% (*Hair et al., 2010*). The items were retained for further analysis if they obtained the ideal communality value of 0.3 and higher and had factor loadings of 0.3 and higher on one factor without extensive cross-loadings on other factors (*Hair et al., 2010*; *Child, 2006*). A low communality value and low factor loadings indicate the failure of items to represent its factor satisfactorily (*Hair et al., 2010*; *Fabrigar & Wegener, 2012*).

### Confirmation of the factor structure and items

Confirmatory factor analysis (CFA) was carried out in the last phase of this study. The CFA was performed using the analysis of moment structure (AMOS) software version 21 (SPSS Inc., Chicago, US) to assess the fitness of the factor structure. Four types of fitness indexes were used to assess the goodness of fit: root mean square error of approximation (RMSEA) of <0.08, comparative fit index (CFI) and Tucker–Lewis fit index (TLI) of >0.90, and chi-square/degrees of freedom (chisq/df) of <3.0 (*Zainudin, 2014*; *Zainudin, 2015*). The factor structure of E-CIS was fit, and the construct validity was achieved, if the model met the fitness indexes' required levels. The modification index (MI) was used to guide model modification because the emphasis on content of the construct above model fit adjustments was necessary, especially when our main research objective was scale development (*Hair et al., 2010*; *DeVellis, 2012*).

### Data and statistical analysis

Frequency or mean was used for descriptive data where appropriate. The construct validity of the new scale was examined firstly by EFA, followed by CFA - as described in above sections. Pairwise deletion for correlations, the listwise exclusion for the EFA, and full estimation maximum likelihood estimation in the CFA were used to handle the missing data. In addition, convergent validity and reliability were also assessed, as described below.

Convergent validity was assessed by computing the value of average variance extracted (AVE) for every construct. The validity was achieved when all of the AVE values exceeded 0.50 (*Zainudin, 2014*; *Zainudin, 2015*). Having less than that would indicate more errors have remained in the items than the variance explained by the latent factor structure (*Hair et al., 2010*). The AVE value was also used to assess the discriminant validity of the new scale. The validity of respective construct was achieved if the square root of its AVE exceeded its correlation value with other constructs in the model (*Zainudin, 2014*; *Zainudin, 2015*).

The new scale's reliability was evaluated using internal consistency and composite reliability (CR), both performed after EFA and CFA. Cronbach's alpha value of 0.70 to 0.90 indicates a satisfactory level of internal consistency (*Zainudin, 2014*; *Zainudin, 2015*) but, for exploratory research such as ours, 0.60 was preferable (*Hair et al., 2010*; *Kline, 2010*). In the EFA, the internal consistency of each factor was examined to improve the total score reliability. An item was to be eliminated if removal of that item would improve item-to-total correlation. A similar method was utilized to assess internal consistency in CFA to ensure a good fit model was obtained. The CR was used to indicate the reliability and internal consistency of a latent construct. The minimum threshold value for CR was
**Table 1  Characteristics participants of the study.**

| Characteristics | N (%) | Mean ± SD |
|---|---|---|
| Age | | 68.64 ± 6.57 |
| 60–69 | 281 (59.8) | |
| 70–79 | 157 (33.4) | |
| 80 and above | 32 (6.8) | |
| Sex | | |
| Men | 219 (46.4) | |
| Women | 251 (53.6) | |
| Educational level | | |
| Never to school | 136 (28.9) | |
| Informal school | 30 (6.4) | |
| Primary school | 158 (33.6) | |
| Secondary school | 134 (28.5) | |
| College/university | 12 (2.6) | |
| Employment status | | |
| Employed | 133 (28.3) | |
| Pensioner | 46 (9.8) | |
| Unemployed | 291 (61.9) | |
| General health status | | |
| Poor to very poor | 65 (13.8) | |
| Moderate | 187 (39.8) | |
| Good to very good | 218 (46.4) | |

**Notes.**
N, number of participants; SD, standard deviation.

0.6, a higher value indicates better reliability (*Zainudin, 2014*; *Zainudin, 2015*; *Fornell & Larcker, 1981*).

## RESULTS

A total of 470 elderly with chronic constipation from various community settings in Kelantan and Terengganu participated in the study. Out of these, 200 of the participants were from Phase 2 of the study and 270 from Phase 3. The age was between 60 and 100 years old, with a mean age of 68.64 ± 6.57. All participants were Muslims with more women than men (53.6 vs. 46.4%). Table 1 shows the background of the participants of this study.

### Validity

From EFA, the initial 40 items were extracted into eight factors and this accounted for 61.0% of the cumulative variance. Subsequently, 27 items were maintained, and 13 items were removed. The Cronbach's alpha for the 27 items was 0.80. The removed items were redundant items that were highly intercorrelated, possess a communality value of less than 0.3, not identifiable with primarily one factor, and the deleted item improved internal consistency reliability (*Child, 2006*; *Hair et al., 2010*). Table 2 shows the remaining items after EFA and the values of its factor loadings, communality, item-total correlation, Cronbach's alpha if an item was deleted, and the Cronbach's alpha of each.

**Table 2  Exploratory factor solution for 27 items.**

| Factor (F) | Items | Factor loading | Communality | Item-total correlation | Cronbach's alpha if item deleted | Cronbach's alpha |
|---|---|---|---|---|---|---|
| | F1a | 0.66 | 0.47 | 0.57 | 0.77 | |
| | F1b | 0.65 | 0.40 | 0.54 | 0.78 | |
| F1 | F1c | 0.55 | 0.40 | 0.53 | 0.78 | 0.80 |
| | F1d | 0.54 | 0.58 | 0.58 | 0.77 | |
| | F1e | 0.59 | 0.49 | 0.53 | 0.78 | |
| | F1f | 0.60 | 0.53 | 0.61 | 0.76 | |
| | F2a | 0.88 | 0.76 | 0.79 | 0.84 | |
| F2 | F2b | 0.82 | 0.69 | 0.76 | 0.85 | 0.88 |
| | F2c | 0.94 | 0.84 | 0.83 | 0.82 | |
| | F2d | 0.49 | 0.59 | 0.63 | 0.90 | |
| | F3a | 0.59 | 0.51 | 0.55 | 0.79 | |
| F3 | F3b | 0.86 | 0.67 | 0.67 | 0.66 | 0.79 |
| | F3c | 0.65 | 0.58 | 0.67 | 0.67 | |
| F4 | F4a | 0.76 | 0.63 | 0.64 | – | 0.78 |
| | F4b | 0.77 | 0.66 | 0.64 | – | |
| | F5a | 0.45 | 0.39 | 0.43 | 0.59 | |
| F5 | F5b | 0.58 | 0.43 | 0.49 | 0.55 | 0.66 |
| | F5c | 0.68 | 0.53 | 0.49 | 0.56 | |
| | F5d | 0.55 | 0.36 | 0.35 | 0.64 | |
| | F6a | 0.54 | 0.48 | 0.52 | 0.72 | |
| F6 | F6b | 0.53 | 0.34 | 0.45 | 0.75 | 0.76 |
| | F6c | 0.58 | 0.57 | 0.56 | 0.69 | |
| | F6d | 0.63 | 0.74 | 0.69 | 0.61 | |
| F7 | F7a | 0.96 | 0.87 | 0.73 | – | 0.84 |
| | F7b | 0.75 | 0.73 | 0.73 | – | |
| F8 | F8a | 0.96 | 0.82 | 0.56 | – | 0.70 |
| | F8b | 0.48 | 0.45 | 0.56 | – | |

**Notes.**
F1, daily activities; F2, treatment satisfaction; F3, lack of control of bodily function; F4, diet restriction; F5, symptom intensity; F6, anxiety; F7, dietary fiber intake; F8, fluid intake.

Application of CFA to establish the fitness of the measurement model obtained from the EFA implied that the first version of E-CIS has a poor fit. This was illustrated by indexes RMSEA = 0.076, CFI = 0.799, TLI = 0.761, and chisq/ $df$ = 2.566. Subsequent modifications to the EFA model resulted in seven factors and 22 items (Fig. 2). Five items were removed due to low factor loadings (0.17 and below). Factors of dietary fiber and fluid intake were justified as a single factor and named as "preventive actions". The revised CFA version was a good model fit, as shown by indexes RMSEA = 0.04, CFI = 0.961, TLI = 0.952, and chisq/ $df$ = 1.44.

The factors and items in the final E-CIS were daily activities (four items), treatment satisfaction (four items), lack of control of bodily function (three items), diet restriction (two items), symptom intensity (two items), anxiety (four items), and preventive actions

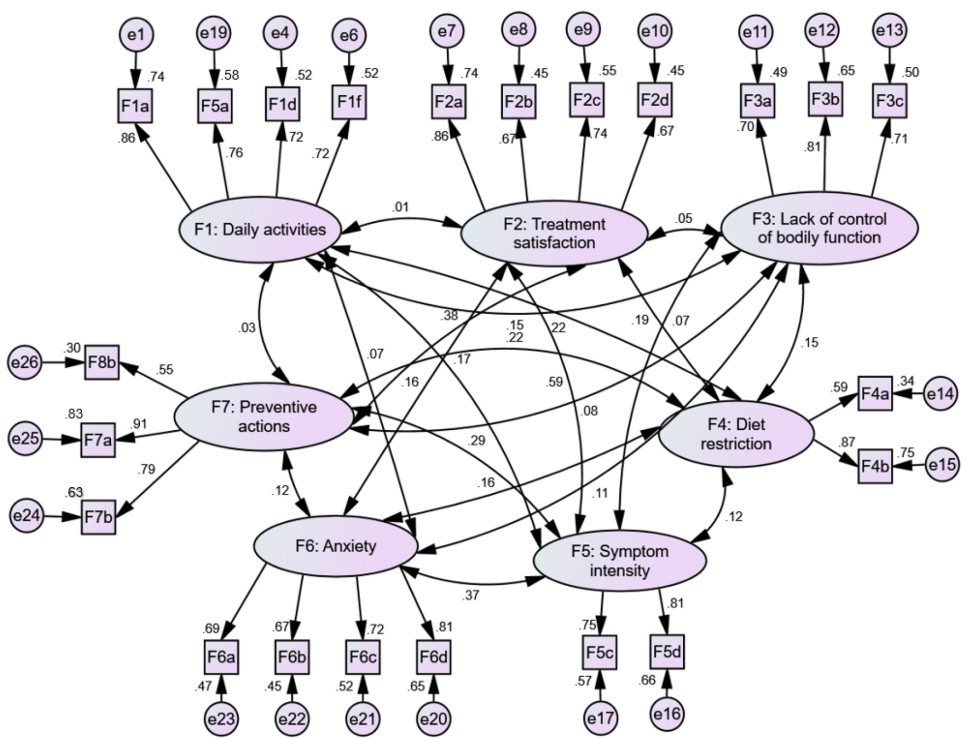

**Figure 2 The measurement model for pooled-CFA.** RMSEA = 0.04, CFI = 0.961, TLI = 0.952, chisq/ $df$ = 1.44.

(three items) (Table 3). The AVE value for all items ranged from 0.53 to 0.61, which exceeded their minimum threshold values. The AVE analysis indicated that all items were statistically significant for the measurement model and were free from redundancy, therefore achieving convergent validity. The square root of AVE for each construct had exceeded its correlation value with the other constructs in the model, thus confirming discriminant validity.

## Reliability

Cronbach's alpha of the final E-CIS was 0.78 and the value for all subscales ranged from 0.66 to 0.85. Specifically, the Cronbach's alpha value for each subscale is as follows: daily activities (0.85), treatment satisfaction (0.82), lack of control of bodily function (0.78), diet restriction (0.66), symptom intensity (0.76), anxiety (0.81) and preventive actions (0.83). The CR values for all the constructs ranged from 0.70 to 0.85. These results indicated that the new scale was accurate and reliable.

## DISCUSSION

Seven factors were identified to have impacted QoL and were further simplified into a 22-item scale, taking into consideration the needs and values of elderly people and their families (*Bakas et al., 2012*). Based on factor analysis, these factors were 'daily activities', 'treatment satisfaction', 'lack of control of bodily function', 'diet restriction', 'symptom

**Table 3  The new version of the Elderly Constipation Impact Scale (E-CIS).**

| Factor/Items | | Factor loading | Cronbach's alpha | CR | AVE |
|---|---|---|---|---|---|
| F1: Daily activities | | | 0.85 | 0.851 | 0.590 |
| F1a | I feel tired after defecation | 0.86 | | | |
| F5a | I feel depressed when I want to defecate | 0.76 | | | |
| F1d | The constipation interrupt my daily activity | 0.72 | | | |
| F1f | I feel interrupted to pray because of the constipation | 0.72 | | | |
| F2: Treatment satisfaction | | | 0.82 | 0.828 | 0.548 |
| F2a | I am satisfied with the effectiveness of the treatment | 0.86 | | | |
| F2b | I am satisfied with the methods used for the constipation treatment | 0.67 | | | |
| F2c | I am satisfied that I get to defecate as I wish for after the treatment | 0.74 | | | |
| F2d | I am confident that constipation will be easy to treat if it recurs | 0.67 | | | |
| F3: Lack of control of bodily function | | | 0.78 | 0.783 | 0.546 |
| F3a | My body feels sick as if I were to have a fever | 0.70 | | | |
| F3b | The effects of my constipation treatment is slow | 0.81 | | | |
| F3c | I feel that I fail to control my constipation | 0.71 | | | |
| F4: Diet restriction | | | 0.66 | 0.699 | 0.546 |
| F4a | Constipation makes me control the quantity of food taken | 0.59 | | | |
| F4b | Constipation makes me choose the type of food taken | 0.87 | | | |
| F5: Symptom intensity | | | 0.76 | 0.761 | 0.614 |
| F5c | I extract the feaces out | 0.75 | | | |
| F5d | I strain strongly during defecation | 0.81 | | | |
| F6: Anxiety | | | 0.81 | 0.815 | 0.525 |
| F6a | I think that my intestinal function has been damaged | 0.69 | | | |
| F6b | I fear that there will be blood coming out from the anus when I want to defecate | 0.67 | | | |
| F6c | I fear that the anus will come out if I strain | 0.72 | | | |
| F6d | I fear that the anus will tear because of the big feaces | 0.81 | | | |
| F7: Preventive actions | | | 0.83 | 0.804 | 0.587 |
| F7a | Constipation makes me take more vegetables in my diet | 0.91 | | | |
| F7b | Constipation makes me take more fruits in my diet | 0.79 | | | |
| F8b | Constipation makes me drink a lot of drinking water | 0.55 | | | |

**Notes.**
F, factor; CR, composite reliability; AVE, average variance extracted.

intensity', 'anxiety', and 'preventive actions'. The factors identified in this study were also similar to the irritable bowel syndrome (IBS)-specific QoL. Included among them were illness experience, stressors, coping mechanisms and psychological state (*Wong & Drossman, 2010*).

In the context of a chronic and complex disorder such as constipation, improvement in health-related QoL is often considered a paramount treatment outcome for the elderly, and therefore it is pertinent to first understand the impact of this disorder (*Pinto Sanchez*

& *Bercik, 2011*; *Belsey et al., 2010*). The current study indicates that the newly developed E-CIS is a valid, reliable and specific tool for elderly people to evaluate the impact of chronic constipation on their QoL.

Although there are several QoL scales for chronic constipation, none fulfilled all the criteria needed to evaluate the impact of this disorder in elderly people (*Marquis et al., 2005*; *Szeinbach et al., 2009*; *Wang et al., 2009*). Firstly, these existing scales were not specifically developed or validated for the elderly population, and it is known that the elderly engage in very different behaviors from the younger population. Secondly, other local or cultural factors such as religion and dietary intake were not included in these scales, but these factors were found to be especially important for the elderly Malaysian population, as reported in a qualitative study (*Patimah et al., 2017*).

The factor of 'preventive action' in E-CIS was important to the elderly - especially the intake of water, fruits, and vegetables. This factor, in addition to factor of 'diet restriction' was a newly identified factor associated with the effect of constipation on QoL. However, previous studies have reported on these factors in other gastrointestinal disorders such as the 'coping/behavior' factor in Fecal Incontinence Quality of Life (FIQoL) (*Hashimoto et al., 2010*; *Kunduru et al., 2015*; *Rockwood et al., 2000*).

'Treatment satisfaction', as factor or an item, is a common inclusion in many constipation-related scales (*Li, Lee & Suen, 2014*; *Müller-lissner et al., 2010*; *Müller-Lissner et al., 2013*; *Nour-Eldein et al., 2014*). Earlier QoL scales had a 'satisfaction' subscale, but items were heterogeneous and based on frequency and regularity in bowel movements, activities, expectations, values, effectiveness and treatment satisfaction (*Marquis et al., 2005*; *Szeinbach et al., 2009*). In newly developed E-CIS, the 'treatment satisfaction' subscale focused on satisfaction with the utilization and effectiveness of an unspecified treatment.

In our study, there are three items reflecting fear in factor 'anxiety'. *Kalat (2014)* had differentiated between fear and anxiety by asserting that fear is an individual's response to an immediate danger, whereas anxiety is their vague sense that 'something bad might happen'. Based on this understanding, we identified that the three items reflected the elderly's fear emotion because the responses were focussed on a specific condition. However, we labelled these items as 'anxiety' factor because of the response to different conditions due to an unknown or poorly defined threat.

In the new scale, the item associated with the effect of constipation on worship activities was identified under the subscale of 'daily activities'. We have shown in our published qualitative study that prayer or worship, a daily practice for many elderly people, was often disturbed because of constipation (*Patimah et al., 2017*). Studies have shown that, in later life, religiosity is a well-documented coping strategy in terms of improvements to psychological well-being (*Momtaz, Hamid & Yahaya, 2009*; *Momtaz et al., 2010*).

The items of this new scale were developed and validated based on a qualitative study using the grounded theory approach (*Patimah et al., 2017*). Besides, the content and face validity of these items were supported by extensive literature reviews and experts' opinions. All the measured items were well loaded to its constructs. This was evidenced by the ideal estimate for standardized loading of 0.7 that was shown for most items in the CFA results (*Hair et al., 2010*). Two items with factor loadings of 0.55 and 0.59 and two items with

R-square of 0.3 and 0.34 were retained. According to the rule of thumb, these values are accepted when the main research objective is scale development (*DeVellis, 2012*; *Hair et al., 2010*). Moreover, the model achieved the required level of the fitness indices and the new 22-item E-CIS has been shown to have good discriminant validity and convergent validity. These findings suggest that the items represent the effect of constipation on the QoL of the elderly satisfactorily.

Calculation of the E-CIS score could be in the form of the total mean score or the total mean subscale scores. Each subscale score provides additional information about individual factors that affect the impact of constipation, compared to a single score. The minimum total score is 20 (very low impact on QoL) and the maximum is 100 (very high impact on QoL). This baseline value is obtained by dividing the minimum and maximum total scores of 22 items by 110, which is the maximum score of the scale. The values are then multiplied by 100 to find the percentage.

The new scale is reliable based on its internal reliability and CR of 0.7 and above (*DeVellis, 2012*). In many studies on scales validation, only a limited number of measures in the construct validity or reliability were reported (*Szeinbach et al., 2009*; *Wang et al., 2009*). In our study, we performed most—if not all—of the validity and reliability assessments, also including EFA and CFA. The reason was that we aimed to have a more comprehensive evaluation of the new scale, but also to have the highest minimum number of items which will suit elderly respondents in our culture better.

Several limitations to this study were identified. The E-CIS was specifically made for elderly people without cognitive impairment - which is a common condition among the elderly. The results on validity and reliability indicated by this scale were not tested with the existing validated and reliable scales, and its sensitivity is not tested by pre and post-treatment. Out of seven factors of the E-CIS, there are two factors, namely 'diet restriction' and 'symptoms intensity' that have only two items. It is generally accepted to have two items per factor or subscale, but the interpretation of results based on the subscale score should be performed cautiously (*Chan et al., 2010*; *Tabachnick & Fidell, 2007*). Other than that, the scale is measured using a 5-point Likert scale to produce the mean score. According to *Chua (2008)*, treating the Likert scale as an interval scale may lead to bias because the mean score tends to accumulate to the middle value. These limitations should be kept in mind when assessing the impact of chronic constipation on the elderly's QoL using this scale.

## CONCLUSIONS

The E-CIS is a constipation-specific QoL scale, which in this study was written in the standard Malay language. This multidimensional scale consists of 22 items with seven subscales. It is proven to be valid and reliable in assessing the impact of chronic constipation on the elderly's QoL among the Malays in Kelantan, regardless of the type of treatment that they have practiced or received. However, further validation study is required in other elderly populations and other ethnic groups to confirm its robustness for multi-ethnic or multinational studies.

## ACKNOWLEDGEMENTS

The authors would like to acknowledge Associate Professor Dr Muhamad Saiful Bahri Yusoff, Ms Fairuza Syahira Zainuddin, and all the experts, who provided constructive comments and insights, and to the research assistant who helped in data collection.

### Funding

This work was supported by the Universiti Sains Malaysia [No. 1001/PPSK/812146]. The funders had no role in study design, data collection and analysis, decision to publish, or preparation of the manuscript.

### Grant Disclosures

The following grant information was disclosed by the authors:
Universiti Sains Malaysia: 1001/PPSK/812146.

### Competing Interests

Yeong Yeh Lee is an Academic Editor for PeerJ.

### Author Contributions

- Abdul Wahab Patimah conceived and designed the experiments, performed the experiments, analyzed the data, prepared figures and/or tables, authored or reviewed drafts of the paper, and approved the final draft.
- Dariah Mohd Yusoff, Azidah Abdul Kadir Siti Hawa Ali, Yeong Yeh Lee, and Yee Cheng Kueh conceived and designed the experiments, analyzed the data, authored or reviewed drafts of the paper, and approved the final draft.

### Human Ethics

The following information was supplied relating to ethical approvals (i.e., approving body and any reference numbers):

The Universiti Sains Malaysia Research Ethics Committee granted ethical approval to carry out the study within its facilities (Ethical Application Ref: USM/JEPeM/272.3. [1.7]).

### Data Availability

The raw data are available in the Supplementary Files.

### Supplemental Information

Supplemental information for this article can be found online at http://dx.doi.org/10.7717/peerj.8581#supplemental-information.

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
