# Peer review of "Psychometric evaluation of a newly developed Elderly-Constipation Impact Scale"

_PeerJ, doi:10.7717/peerj.8581_

## Round 0.1 · original submission · Major Revisions

Your manuscript has been reviewed and requires modifications prior to making a decision. The comments of the reviewers are included at the bottom of this letter. The reviewers think that your manuscript needs revisions on citing references in text, details of sample selection, figures and tables. Reviewer 1 indicated that it would be better to have a section in the method about sample and data assessment. In addition to this, you should update your figures and tables. Reviewer 2 commented on the title of the manuscript, the results section of the study, and had concerns on the sample. I agree with this evaluation and I would, therefore, request for the manuscript to be revised accordingly. I would also like to ask you the following questions:

• Why did you prefer to use Promax rotation rather than Varimax rotation?
• How did you determine the sample size for this study? What was the sampling method of the study? For example, did you use cluster sampling or stratified sampling or etc?

·

Basic reporting

This is a well written paper with a language that makes it easy to follow. The concept of Quality of Life is addressed in the introduction, as well as in terh aim of the abstract and the conclusions. However, the connection to the concept apart from that is not explicit. If QoL is supposed to be in focus as an outcoem measurment, it needs to be described and explained in further details in order for the reader to understand the concept better here.
The structure, with the use of figures and tables are relevant (se comments below) and the data is shared.

Experimental design

¤ The aim was to develop and validate a culturally sensitive scale. To me, this implies that the scale should be cultural sensitive in a more general sense. However, from my reading, I understand that the authors thought that already existing scales were mostly developed in Western countries. Consequently regarded unsuitable for a population from Asian countries, and therefore the research project developed a scale more suitable for an Asian population, more specifically a Malaysian population. To me, then the scale is perhaps not culturally sensitive in a broad sense, but more restricted to a certain population, and still could be regarded as a culturally specific scale, but for another population. What do you think about this?

¤ Quality of Life (QoL) is addressed in the introduction, and seems then as a focus for the impact of constipation. However, the connection is not so explicitly visible in the rest of the manuscript, or in the instrument variables. It seem as if the impact is more on daily life in general than on QoL. Some of the items are more about the constipation itself or the prevention and management of it, and not on the impact on elderly people. The connection to QoL is in a way mostly implicit. If QoL is to be the core outcome, the concept is in need of some further description in the background, perhaps defining the understanding of the concept in order for the reader to understand the findings.

¤ The aim does not focus on QoL either, it is described in a much broader sense as “impact of chronic constipation on elderly people”. However, it is stated slightly different in the abstract: “effect of chronic constipation on quality of life among the elderly”. I suggest that the aim should be stated exactly the same in both places.

¤ The sampling procedure of the participating 470 elderly persons need to be described in some more details. They came from community settings, but from how many settings, and how they were invited is not described in details. Further, how many elderly persons were approached and assessed regarding chronic constipation to end up with the 470. Now some information is provided under the heading item generation, but it would be better to have a section in the method about sample and data assessment.

Validity of the findings

No comment

Additional comments

¤ The labelling of the factors could be discussed in some cases. Examples given, there are three items reflecting fear in the factor Anxiety. Fear and anxiety are different phenomena. Fear is related to a known or understood threat, whereas anxiety is a reaction followed from an unknown, expected or poorly defined threat. Also the first item in the factor is more about fear than anxiety since it is so focused. TI suggest that you think through the labeling of the factors one more time.
¤ Provide a reference to figure 1 and 2 in the main text to direct the reader.

¤ Figure 1. Why do you not describe the whole process description by adding the resulting restructruing after the CFA, ending up with 22 items in 7 factors? In a way, it depends on where you plan to have the figure in the text and if you "only" want it to describe the process up to the final analytic stage. Consider the suggestion, it gives a nice overview for the reader. Also, it could benefit from being a bit smaller in size.

¤ Table 2 Preferably you should have presented how each item loaded on the different factors in the exploratory factor analysis., not just for the one where you placed it. A suggestion, flip the columns and rows displayng the items on the rows sorted by the factors been identified and have columns, first for each factor, followed by communalities and Cronbach's alpha values for the factor. You can address in the text that all those variables had communality values over 0.30

¤ Table 1. A minor detail: the %-figures for sex do not match on the first decimal when you compare the table and the text on row 222.

¤ Figure 2.
- Are all information in the right place? Did not all items have measurement errors? E.g. item r2-values for e19/F5a, e24/F7b and e25/F7a are either missing or non-existent.
- Further, the factor loadings display two pieces of information for F1d, F7b and F7b which makes me wonder if some of those figures are representing item r2-values instead.
- Could the factor labels be displayed in the figure so that the figure can stand on its own?
¤ A discussion could have been held about the fact that some items were kept in the model despite low factor loadings (<0.6) or low r2-values (<0.4). I guess it is because the fitness indices for the model already achieved a required level.

Reviewer 2 ·

Basic reporting

The manuscript is well written and with a clear English.
I suggest to authors to improve the title of the manuscript. The factor structure of an instrument is only a part of the instrument development. To offer readers a wider comprehension of the scientific work presented in the manuscript, I suggest to authors to include in the title words such as "psychometric evaluation" or "development".
The article structure is suited to journal recommendations. I have a few concerns about results' session: it is only 34 raw long and its quite short when compared to other parts of the manuscript. My suggestion is to explain more how and why the authors removed 13 items.
The references are up to date and pertinent to the context described. Some citations are not respecting the journal norms, therefore authors should be careful to follow the chronological order of references in the manuscript text.

Experimental design

The research question and the methods session of the manuscript are described in details.
I raise only a few questions to the authors.
The instrument is addressed to the elderly, but your sample includes people aged 60 and above (raw 128). The international literature defines "aged" people starting from 65 years old and in some cases starting from 75 years old. The authors need to explain why they made such a choice.
On raw 139-140 the authors state :"The aim was to involve a heterogeneous population that allowed for the detection of differences in scores across the new scale". In the manuscript they do not present any data about the construct validity of the instrument. I should expect some initial hypothesis like: people referring slightly severe constipation might present lower scares on the scale while people presenting severe constipation might have higher scores. Further, on the results session I expected to find information about the construct validity of the instrument.
It is not quite clear to me why authors have chosen the sample characteristics presented in Table 1, for example educations, employment and if there is any relation between these factors and the constipation. I suggest to include in the methods session some more explanations about this. Moreover, how do authors measure general health status?

Validity of the findings

The findings are interesting and well described and discussed. I have no further comments on it.

---

## Round 0.2 · accepted · Accept

The authors addressed the reviewers' concerns and substantially improved the content of the manuscript. So, based on my own assessment as an academic editor, no further revisions are required and the manuscript can be accepted in its current form.

·

Basic reporting

The manuscript has now been developed according to previous given comments from reviewers and editor.

Experimental design

No further comments

Validity of the findings

No further comments

Additional comments

No further comments

Reviewer 2 ·

Basic reporting

no comment

Experimental design

no comment

Validity of the findings

no comment

Additional comments

The authors accepted the suggestions of the reviewers and now the manuscript results more fluid.